# Recent Advances in the Strategies of Simultaneous Enzyme Immobilization Accompanied by Nanocarrier Synthesis

**Xinrui Hao, Pengfu Liu and Xiaohe Chu \***

Collaborative Innovation Center of Yangtze River Delta Region Green Pharmaceuticals, Zhejiang University of Technology, Hangzhou 310014, China; 15858483966@163.com (X.H.); liupengfu@zjut.edu.cn (P.L.)
\* Correspondence: chuxhe@zjut.edu.cn

**Abstract:** In recent years, with advancements in nanotechnology and materials science, new enzyme immobilization strategies based on nanomaterials have continuously emerged. These strategies have shown significant effects on enhancing enzyme catalytic performance and stability due to their high surface area, good chemical stability, and ease of enzyme binding, demonstrating tremendous potential for industrial applications. Those methods that can rapidly synthesize nanocarriers under mild conditions allow for the one-step synthesis of nanocarriers and enzyme complexes, thereby exhibiting advantages such as simplicity of process, minimal enzyme damage, short processing times, and environmental friendliness. This paper provides an overview of simultaneous enzyme immobilization strategies accompanied by nanocarrier synthesis, including organic–inorganic hybrid nano-flowers (HNFs), metal–organic frameworks (MOFs), and conductive polymers (CPs). It covers their preparation principles, post-immobilization performance, applications, and existing challenges.

**Keywords:** enzyme; immobilization; nanomaterial; organic–inorganic hybrid nanoflowers (HNFs); metal–organic frameworks (MOFs); conductive polymers (CPs)



## 1. Introduction

Enzymes are naturally evolved, biodegradable, and biocompatible, highly efficient catalysts widely applied in various scientific, technological, and industrial fields due to their notable catalytic efficiency, substrate specificity, and environmentally friendly nature under mild reaction conditions [1]. Despite the advantages of enzymes over chemical catalysts, such as high catalytic efficiency, high substrate specificity, minimal side reactions, and environmental friendliness, the inherent protein nature of enzymes makes them prone to structural collapse and loss of activity under extreme environmental conditions [2]. Additionally, enzymes typically exist in a soluble state within the reaction system, making it challenging to separate and extract the products after the reaction, leading to difficulties in product isolation and enzyme recovery, consequently increasing production costs. These issues constrain the industrial application of enzymes [3].

Enzyme immobilization technology refers to the process of fixing enzymes within a defined spatial range using physical or chemical means to protect them from environmental factors and enhance their stability [4]. Common enzyme immobilization methods include adsorption [5–9], encapsulation [10–14], and cross-linking [15–19]. Apart from the impact of immobilization methods, the choice of immobilization carriers also significantly influences the catalytic performance of enzymes [20]. Since the first report of enzyme immobilization in 1916, many carriers [21], such as chitosan, oxidized graphene, or polyurethane foam, have been widely used for enzyme immobilization. However, the development of novel immobilization matrices and techniques continues to attract significant attention, including polymer matrices, nanomaterials, porous materials, microcapsules, and magnetic materials [22]. With the vigorous development of nanotechnology, the application of nanomaterials in the field of enzyme immobilization has garnered widespread attention. This is

due to the unique properties of nanomaterials [23], such as a larger surface area-to-volume ratio, greater resistance to stress, and lower mass transfer resistance, which help address the drawbacks of traditional enzyme immobilization methods.

Currently, there are many strategies [24,25] for enzyme immobilization based on nanomaterials. However, most of these immobilization strategies typically involve the preparation of carrier materials for enzyme loading, followed by the physical or chemical binding of the carrier and enzyme. The preparation and separation of carriers involve multiple steps and cumbersome processes, which are time-consuming [26]. Therefore, there is a growing research interest in developing strategies that can simultaneously synthesize carrier materials and immobilize enzymes. These immobilization strategies involving nanocarriers can be rapidly synthesized through simple physical or chemical reactions, with characteristics such as easy preparation steps, mild conditions, short processing times, and minimal enzyme damage often achieving better results. In this review, we have selected three immobilization strategies that align with these characteristics, including organic–inorganic nano-flowers, metal–organic frameworks, and conductive polymers as nanocarriers for enzyme immobilization (Figure 1). We have outlined their mechanisms of action, general operational processes, key influencing factors, and post-immobilization effects. Additionally, we have briefly discussed the existing limitations and improvement directions for these three strategies. This paper aims to provide more theoretical references for the research and development of strategies that simultaneously synthesize carrier materials and immobilize enzymes during the carrier synthesis process. Due to the abundance of acronyms in this review, we have created an acronym table for better understanding (Abbreviations).

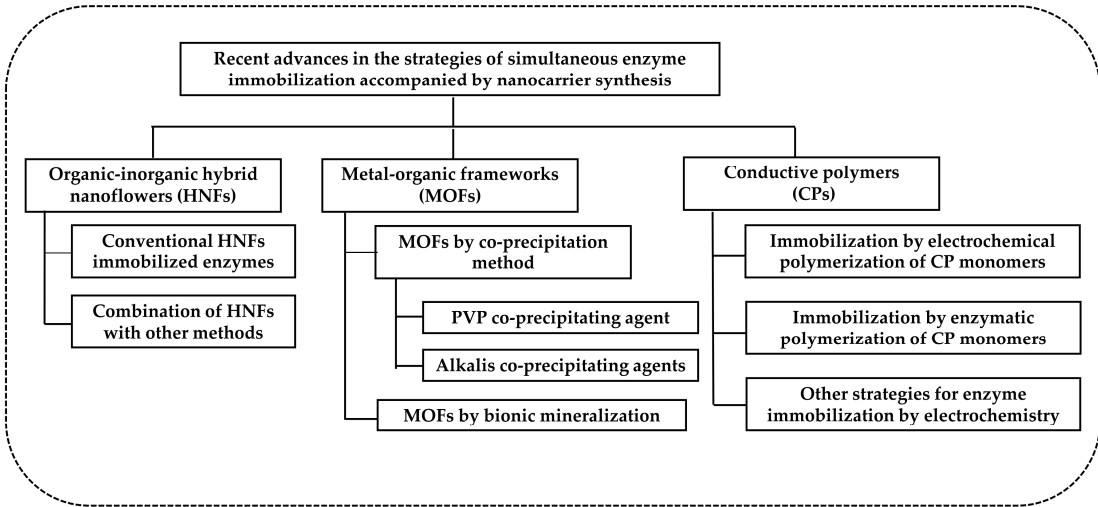

**Figure 1.** A flowchart for the generalized classification of HNFs, MOFs, and CPs as nanocarriers for enzyme immobilization.

## 2. Organic–Inorganic Hybrid Nanoflowers

Nanoflowers refer to nanostructures characterized by a distinctive flower-like morphology at the nanometer scale. These structures have undergone extensive research owing to their promising applications in diverse fields, including catalysis, energy storage and conversion, sensors, and biomedical imaging. The notable characteristics of nanoflowers, such as their high surface area and unique morphology, contribute to enhanced catalytic activity, superior optical properties, and increased sensitivity. Moreover, the flower-like structure offers a larger contact area for biomolecules to bind, making nanoflowers particularly suitable for applications in biosensors and bioimaging. Based on differences in composition, nanoflowers can be classified into several types, including inorganic nanoflowers [27–30], organic nanoflowers [31–33], and hybrid nanoflowers (consisting of both organic and inorganic components) [28]. In 2012, Ge et al. [34] first discovered the

formation of organic–inorganic hybrid nanoflowers by co-incubating proteins, copper ions, and phosphates. During the synthesis process, inorganic components precipitate, grow, and aggregate around the enzyme, eventually forming flower-shaped particles with a size of approximately 100–500 nm. This novel nanomaterial is referred to as organic–inorganic hybrid nanoflowers (HNFs) [35].

Subsequent research has revealed that HNFs constitute a versatile strategy for the immobilization of large biomolecules. Antibodies [36–40], proteins [41], DNA [42–46], amino acids, and other biomolecules can serve as organic components of HNFs. In principle, any biomolecule with metal-binding sites can form complexes with metal ions through coordination [47]. HNFs integrate the advantages of both organic and inorganic materials, making them an ideal carrier for enzyme immobilization. The formation of HNFs with enzymes as organic components generally involves three steps: coordination, precipitation, and self-assembly [2,35,48]. In the initial stage, metal ions react with phosphate ions to form primary crystals of metal phosphates. Functional groups on proteins (amide, carboxyl, or hydroxyl groups) coordinate with metal ions, providing a site for the nucleation of proteins and primary crystals. In the second stage, metal phosphate crystals begin to grow at the metal ion-binding sites and continue to grow through the continuous binding of protein nanoflower petals and primary crystals. In the final stage, the formation of nanoflowers is completed through anisotropic growth. In recent years, HNFs have garnered widespread attention due to their comprehensive functionality, combining organic and inorganic materials, as well as their environmentally friendly and straightforward preparation steps.

*2.1. Conventional HNFs Immobilized Enzymes*

HNFs can be obtained by mixing metal ions, phosphate, and enzyme solutions and allowing them to co-incubate. Among the various enzyme nanoflowers studied, there has been a significant focus on single-enzyme HNFs, with enzymes such as laccase [49], lipase [50], horseradish peroxidase (HRP) [51], glucose oxidase (GOx) [52], $\alpha$-amylase [53], urease (Ur) [54], and papain [55] being successfully prepared in the form of single-enzyme HNFs [3] (see Table 1).

**Table 1.** Summary of HNF applications and enhanced performance.

| Enzyme | Metal Ions | Applications | Improved Performance | Ref. |
|---|---|---|---|---|
| $\omega$-Transaminase | $Co^{2+}$ | Production of chiral amines | Enhanced reusability | [56] |
| lipase | $Zn^{2+}$ | Regioselective acylation of arbutin | Enhanced reusability | [57] |
| L-arabinose isomerase | $Mn^{2+}$ | Synthesis of *D*-tagatose | Enhanced reusability and storage stability | [58] |
| Lactoperoxidase | $Cu^{2+}$ | Detection of dopamine and epinephrine | Enhanced activity, pH stability, and reusability | [59] |
| Lipase from thermomyces lanuginosus | $Ca^{2+}$ | Proof of concept | 21.7 times more catalytic activity and thermal stability than a free enzyme | [60] |
| Phosphotriesterase | $Co^{2+}$ & $Mn^{2+}$ | Use in nerve agent (GD and VX) degradation | Enhanced stability and reusability | [61] |
| Burkholderia cepacia | $Ag^+/Fe^{2+}/Cu^{2+}/Au^{3+}$ | Proof of concept | Enhanced stability | [62] |
| Aldehyde/ketone reductase and alcoholdehydrogenase | $Ca^{2+}$ | Synthesis of (*S*)-1-(2,6-dichloro-3-fluorophenyl) ethyl alcohol | Enhanced thermal stability | [63] |

**Table 1.** *Cont.*

| Enzyme | Metal Ions | Applications | Improved Performance | Ref. |
|---|---|---|---|---|
| Polyketone reductase and glucose dehydrogenase | $Ca^{2+}$ | Synthesize (R)-(-)-pantolactone | Enhanced stability and reusability | [64] |
| Galactose oxidase and horseradish peroxidase | $Mn^{2+}$ | Detection of glutamic acid | Enhanced reusability | [65] |
| Horseradish peroxidase and glucose oxidase | $Cu^{2+}$ | Degradation of acridine and wastewater treatment | Enhanced pH stability | [66] |
| Nucleoside kinase and polyphosphate kinase | $Cu^{2+}$ | Generation of nucleotides | Enhanced reusability | [67] |
| Glucose oxidase and lipase | $Cu^{2+}$ | Epoxidation of alkenes | Enhanced reusability | [68] |
| Streptavidin and horseradish peroxidase | $Cu^{2+}$ | Colorimetric sensor for alpha-fetoprotein (AFP) detection | Enhanced storage stability | [69] |
| Glucose oxidase and horseradish peroxidase | $Cu^{2+}$ | Counting the number of living bacteria in urine | Enhanced thermostability | [70] |

In the majority of reported cases, hybrid nanoflowers (HNFs) exhibit higher catalytic activity compared to the corresponding free enzymes. This phenomenon can be attributed to the following reasons [2,15,50]: (1) Increased surface area: The larger surface area of nanoflowers facilitates the enrichment of substrates around the enzyme, thereby enhancing the catalytic efficiency. (2) Metal Activation: Certain metal enzymes can be activated by the metal ions present in the nanoflowers. (3) Conformational favorability: Enzymes immobilized through coordination bonds in nanoflowers can maintain a conformation favorable for catalysis. The enhanced stability observed in HNFs is attributed to the interaction between the rigid inorganic precipitate and the flexible enzyme. This interaction provides a rigid framework for the enzyme, reducing the degree of conformational change. As a result, it prevents enzyme denaturation under extreme conditions, improving the stability of the enzyme in complex environments and under various conditions [60,71].

In practical production processes, single-enzyme HNFs may not be suitable when multiple enzymes are involved in a reaction. Single-enzyme HNFs struggle to rapidly remove unstable intermediates, hindering the progress of the reaction. In contrast, multi-enzyme HNFs, based on the simultaneous immobilization of multiple enzymes, prove more effective. The co-immobilization of enzymes not only reduces the cost of repetitive operations but also brings enzymes closer together. This proximity facilitates the transfer of intermediates between different enzymes in cascade reaction systems [72,73]. Han et al. reported the co-immobilization of cellulase, endoglucanase, and β-glucosidase to prepare multi-enzyme HNFs (ECG-NFs) for the one-pot conversion of cellulose to glucose. ECG-NFs, compared to free multi-enzyme systems, exhibited excellent performance in terms of pH stability, thermal stability, storage stability, and catalytic efficiency [73]. Aydemir et al. synthesized TrpE@ihNFs using α-amylase, lipase, protease, and $Cu^{2+}$ ions as raw materials. TrpE@ihNFs demonstrated significantly higher enzyme activity and stability than other free enzymes, protecting the enzymes from the impact of extreme temperature and pH fluctuations. These multi-enzyme HNFs show promise in applications such as wastewater treatment, biosensors, biocatalysts, and future bio-related devices [74].

### 2.2. Factors Affecting the Formation of HNFs

HNFs' morphology and performance are influenced by factors such as temperature, pH, time, metal ions, and enzyme concentration during the preparation process. Therefore, when using this method to immobilize enzymes, it is generally necessary to systematically optimize these conditions to achieve the best results. However, there are relatively few reports on strategies for the controlled synthesis of HNFs. Wang and colleagues reported

the preparation of a series of HNFs [chloroperoxidase (CPO)-(Cu/Co/Cd))$_3$(PO$_4$)$_2$] and their application in crystal violet decolorization (Figure 2). They introduced an excess of chloride ions during the preparation of HNFs to form [MCl$_4$]$^{2-}$ complexes, which slowed down the precipitation of phosphates and simultaneously promoted the coordination of M$^{2+}$ with amide groups. This affected the growth process and morphology of the nanoflower [75].

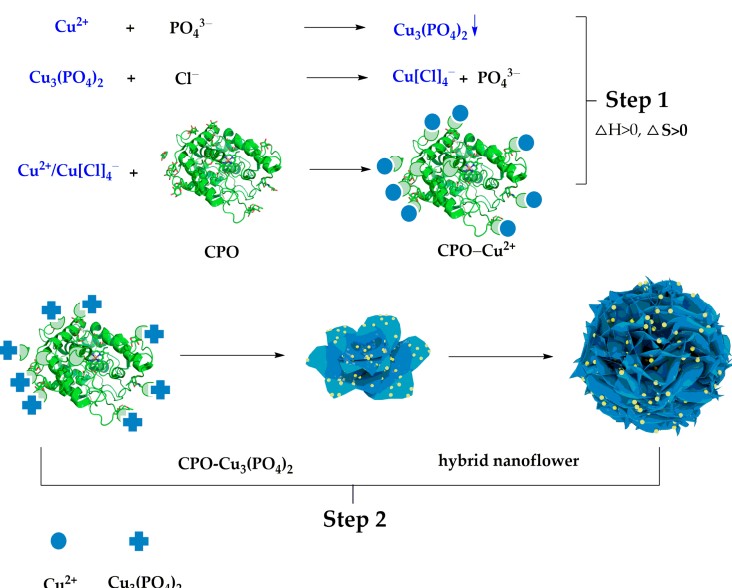

**Figure 2.** The growth mechanism of hybrid materials [75].

Hua and colleagues reported a method for preparing size-controlled laccase-copper phosphate composite material (Lac@Cu$_3$(PO$_4$)$_2$) (Figure 3). using ethylenediaminetetraacetic acid (EDTA) as a chelating compound. Lac@Cu$_3$(PO$_4$)$_2$ was employed for the rapid and sensitive detection of phenol in water, exhibiting excellent catalytic activity and reusability. In terms of activity, Lac@Cu$_3$(PO$_4$)$_2$ was 7–11 times higher than free laccase [76].

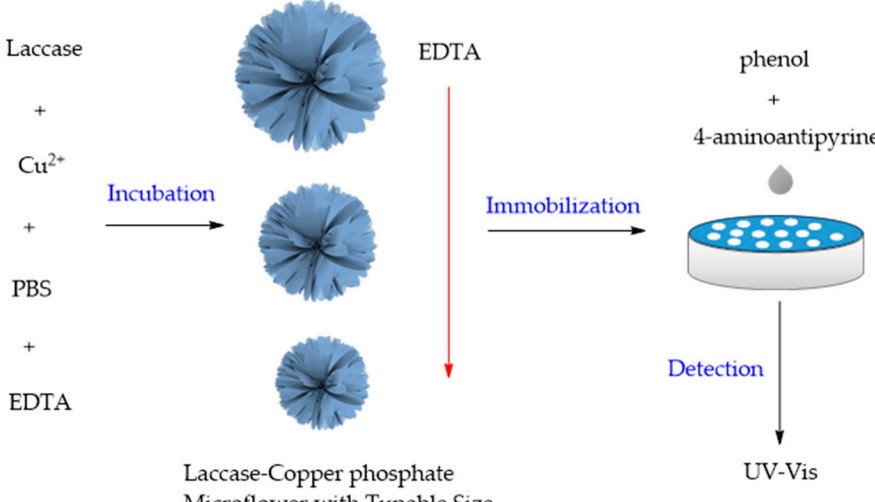

**Figure 3.** Controlled synthesis of the 3D flower-like laccase copper phosphate composites with tunable sizes in the presence of different concentrations of EDTA and the detection of phenol by the laccase microflowers immobilized on membrane [76].

The reusability of HNFs has been shown to be improved by changing the type of metal ions. Patel and colleagues immobilized laccase using copper (Cu) and zinc (Zn) ions in a

phosphate buffer solution. The hybrid Cu/Zn ion HNFs formed a novel multi-metal HNF system, Cu/Zn Lac, showing higher catalytic activity and reusability. The $k_{cat}/K_m$ was enhanced 3.2-fold through the multimetal hybrid Cu/Zn-Lac compared with values of 2.1- and 2.4-fold with Cu-Lac and Zn-Lac over the free enzyme (71.0 s$^{-1}$ μM$^{-1}$). The results indicate that multi-metal HNFs are more beneficial than single-metal HNFs and can be used to immobilize various enzymes for sustainable applications [77].

While the activity and stability of HNFs have significantly improved, their synthesis speed is generally slow, often requiring incubation at room temperature for up to 1–3 days [78]. This severely limits their practical applications. To address this issue, Batule and colleagues developed an ultrasonic-assisted method for synthesizing flower-shaped HNFs. This method is simple and efficient, enabling the synthesis of HNFs using laccase as a model protein and $Cu_3(PO_4)_2$ within 5 min at room temperature. The resulting laccase nanoflowers exhibited remarkable enhancements in activity, stability, and reusability [79].

### 2.3. Combination of Hybridized HNFs with Other Immobilization Methods

Despite the many advantages of HNFs, their structure, characterized by numerous fragile petal-like structures, is prone to breakage during processes such as stirring or centrifugation, leading to structural fragility [80]. This mechanical instability limits their reusability and application in the field of biocatalysis [81]. Scientists have addressed this issue by enhancing the mechanical strength of HNFs through the introduction of connections between petals, resulting in the synthesis of efficient, durable, and recyclable HNFs.

Natural biopolymers such as gelatin (Ge) and chitosan (CS) possess an ample number of amino and hydroxyl groups, serving as nucleation points for crystal formation. They reduce the nucleation barrier of crystal formation and promote the heterogeneous nucleation process of inorganic minerals [82], achieving controlled mineralization. The regulation of biomineralization by natural biopolymers represents a promising immobilization method characterized by excellent stability and recyclability. Xu et al. utilized the natural biopolymer CS to regulate the biomimetic mineralization of calcium phosphate (CaP). Through self-assembly, they immobilized sucrose phosphorylase (SPase), creating a novel type of HNF with excellent stability and catalytic activity: CS-CaP@SPase (Figure 4). Even after 10 cycles, the relative activity of CS-CaP@SPase remained around 80%, and after 15 days, it maintained a relative activity of approximately 75% [82].

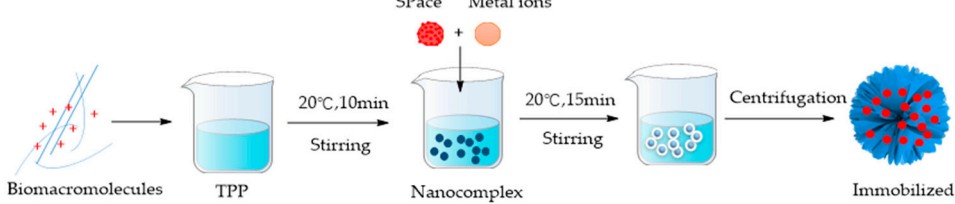

**Figure 4.** Schematic illustration of the preparation process of the CS-CaP@SPase [82].

The incorporation of carbon nanotubes (CNTs) into HNFs can enhance the mechanical strength of the nanoflowers and provide additional accessible adsorption binding sites, effectively improving enzyme activity. Dadi et al. synthesized HRP-NF@CNTs composed of horseradish peroxidase (HRP), $Cu^{2+}$, and CNTs using an in situ method. Due to the synergistic integration of CNTs and $Cu_3(PO_4)_2$ crystals, HRP-NF@CNTs exhibited outstanding peroxidase-like activity and stability. Compared to HRP-NF, HRP-NF@CNTs also demonstrated excellent stability after 10 cycles. Furthermore, compared to free HRP, the kinetic parameters of HRP-NF@CNTs were improved, with a reduction in the Km value by 18.05 times [83].

Cheno et al. coupled amine-functionalized $Fe_3O_4$ magnetic nanoparticles (MNPs) with glucose oxidase (GOx) molecules, followed by the addition of $Cu_3(PO_4)_2$ to prepare embedded MNPs-GOx NFs. MNPs-GOx NFs exhibited excellent peroxidase-like activity and demonstrated more sensitive glucose detection activity compared to MNPs mixed

with free GOx. The close proximity between MNPs and $Cu_3(PO_4)_2$ crystals facilitated the accumulation of $H_2O_2$ molecules produced during the catalysis of glucose oxidation by GOx, creating a substrate channel environment that enhanced catalytic activity. MNPs-GOx NFs prepared through this strategy demonstrated excellent selectivity, stability, and magnetic reusability [84].

For cofactor-dependent reactions, the recycling of expensive cofactors is crucial for implementing enzyme catalysis at an industrial scale. The immobilization of cofactors is considered an effective strategy to address these challenges. Cao and colleagues pioneered the co-immobilization of omega-transaminase ($\omega$-ta) and 5′-pyridoxal phosphate (PLP) within $Co_3(PO_4)_2$ nanoflowers, creating a novel self-sufficient biocatalyst, $\omega$-TA-PLP@$Co_3(PO_4)_2$. In comparison to free $\omega$-ta, $\omega$-TA-PLP@$Co_3(PO_4)_2$ demonstrated higher catalytic efficiency, stability, storage stability, and repeatability [56].

In recent years, due to their larger surface area, greater stress resistance, and reduced mass transfer resistance, HNFs have exhibited higher activity, stability, and reusability compared to free enzymes. Their applications in sensor devices [85], biocatalysis [47,86], biomedical [87], and other fields have become attractive research areas. However, there are still some issues with this technology that require further investigation. On one hand, not all metal ions have a promoting effect on enzymes during the immobilization process, indicating the need for detailed studies on the enzyme's properties and metal ions before adopting this technology for immobilization. On the other hand, in the study of the simultaneous immobilization of multiple enzymes, the order and ratio of immobilizing different enzymes can have varied effects on the overall catalytic performance and properties of the enzymes. This necessitates comprehensive exploration and rational design in the process of preparing HNFs. In conclusion, future efforts should focus on developing more efficient HNFs, which will have a profound impact on areas such as sensors, biomedicine, and detection.

## 3. Metal–Organic Frameworks

Metal–organic frameworks (MOFs), also known as porous coordination polymers [88], are multi-dimensional lattices [89] self-assembled from metal ions serving as central ions, inorganic metal ions, or metal ion clusters, and organic ligands connected through coordination bonds [90]. The principle of MOF precipitation primarily involves the formation of coordination compounds between metal ions and organic ligands through coordination bonds. These coordination compounds begin to aggregate when they reach a certain concentration, forming small aggregates known as nuclei. After the formation of MOF nuclei, monomers of MOF in the surrounding solution diffuse to the nuclei and undergo coordination reactions, causing the nuclei to gradually grow and ultimately form granular solid-phase products. The metal ions or clusters [91–94] within MOFs are predominantly derived from transition elements or lanthanides found in the periodic table [22]. The organic ligands in MOFs mainly consist of polyacidic ligands, poly-pyridine-type ligands, and functional group hybrid ligands. Owing to the multifunctionality of metal nodes and ligands, along with their rich geometric shapes and diverse connectivity, MOFs possess structural flexibility, tunable nanospace, and controllable synthesis advantages.

In recent years, MOFs as enzyme immobilization carriers have been widely investigated. Currently, methods for in situ enzyme immobilization using MOFs mainly include the bottom-up synthesis approach [95,96]. This method primarily involves using enzyme molecules as cores to induce the formation of the MOF framework, allowing the in situ growth of MOF crystals around them, ultimately forming a highly ordered and stable MOF shell layer around the enzyme molecules [97]. This achieves the in situ immobilization of enzyme molecules within the MOF crystals [22]. In this method, the required structural properties of MOFs, such as surface area, pore size, shape, and pore volume, can be easily controlled [98], and the presence of the MOF shell shields the enzyme from the impact of extreme environments, improving the enzyme's storage stability and tolerance to temperature, pH, and organic solvents [99–101] (refer to Table 2). The immobilization

of enzymes with MOFs enhances stability, possibly for the following reasons [102]: (1) enzymes are fixed in the pores through various interactions, tending to maintain their active conformation, aiding in increased stability and reduced enzyme leakage during repeated use; (2) the appropriate pore structure in MOFs can provide size selectivity for specific substrates, offering an additional protective layer for the enzyme, and substrates must diffuse through the pore channels to approach the enzyme, positively affecting the catalytic process; (3) MOFs prevent enzyme aggregation through physical isolation. The bottom-up synthesis approach, also known as in situ encapsulation, can be further categorized into co-precipitation and biomimetic mineralization methods based on whether additional co-precipitants are needed to form immobilized enzymes [100].

**Table 2.** Summary of the enzyme–MOF composite's one-step synthesis.

| MOF | Metal Ions | Enzyme | Applications | Improved Performance | Ref. |
|---|---|---|---|---|---|
| ZIF-8 | $Zn^{2+}$ | Cytochrome C | Oxidation of Amplex red | Enhanced activity | [103] |
| ZIF-8 | $Zn^{2+}$ | Horseradish Peroxidase and Glucose Oxidase | Selective glucose detection | Enhanced activity and selectivity | [104] |
| ZIF-8 | $Zn^{2+}$ | Carbonic Anhydrase | $CO_2/N_2$ selectivity composite membranes | Enhanced stability | [105] |
| ZIF-8 | $Zn^{2+}$ | Horseradish Peroxidase | Proof of concept | Enhanced thermal stability | [106] |
| ZIF-8 | $Zn^{2+}$ | Lipase QLM | Kinetic resolution of (*R*, *S*)-2-octanol | Enhanced activity and reusability | [107] |
| ZIF-90 | $Zn^{2+}$ | Catalase | Biocatalysis | Enhanced activity | [108] |
| MIL-53/ NH2-MIL-53 | $Al^{3+}$ | β-glucosidase/ Laccase | Proof of concept | Enhanced organic solvent stability | [109] |
| MIL-88A | $Fe^{3+}$ | Glucose Dehydrogenase/ Horseradish Peroxidase /Acetylcholinesterase | Proof of concept | Enhanced reusability | [110] |
| MIL-100 | $Fe^{3+}$ | Lipase PPL | Synthesis of benzyl cinnamate | Enhanced thermal, pH, and stability | [111] |
| Fe-MOF | $Fe^{3+}$ | Alcoholdehydrogenase/ Lipase/Glucose Oxidase | Biocatalysis | Enhanced reusability | [112] |

### 3.1. MOFs by the Co-Precipitation Method

The co-precipitation method accomplishes the synthesis of MOFs and the encapsulation of enzymes in a single step, making MOFs with smaller pore sizes than enzymes suitable as encapsulation carriers. It is noteworthy that the crystals formed by the co-precipitation method (120 nm) have smaller sizes compared to the biomimetic mineralization method (500 nm) [96]. This results in a more uniform distribution of enzymes inside the MOFs, facilitating contact with reactants and consequently enhancing the rate of enzyme-catalyzed reactions. Co-precipitants are typically substances that increase the concentration of metal cations near the enzyme surface or organic ligands, promoting MOF nucleation. In the in situ enzyme immobilization strategy, commonly used co-precipitants to promote MOF formation include polyvinylpyrrolidone (PVP) and alkalis (such as NaOH, $NH_4OH$, etc.).

#### 3.1.1. PVP Co-Precipitating Agent

In the process of encapsulating enzymes into MOFs, PVP is commonly used as a co-precipitant to facilitate the formation of complexes. PVP aids in maintaining the dispersion of enzymes during immobilization and stabilizes enzymes in the solution through electrostatic/hydrogen bonding interactions [22,108]. Simultaneously, PVP promotes MOF nucleation through its weak coordination affinity for metal cations via the pyrrolidone

groups. Lyu and colleagues reported a straightforward method for the direct synthesis of biologically active protein-encapsulated MOFs (Figure 5). They achieved the direct embedding of cytochrome c (Cyt c) into zeolitic imidazolate framework-8 (ZIF-8) by adding a solution containing Cyt c and PVP to a methanol solution of zinc nitrate hexahydrate and 2-methylimidazole (mIM). They find that the embedded Cyt c shows a significantly decreased $K_m$ of $H_2O_2$, which suggests the possibility that the immobilized Cyt c has a higher substrate affinity toward $H_2O_2$ molecules. The Cyt c embedded in ZIF-8 displayed a tenfold increase in peroxidase activity compared to free Cyt c in solution [103].

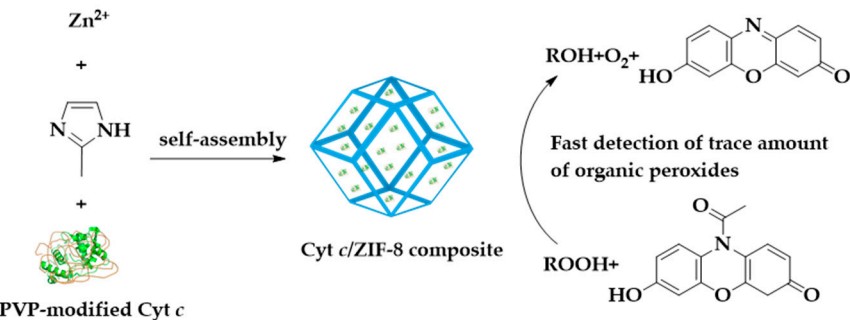

**Figure 5.** Preparation of the cytochrome c-embedded ZIF-8 [103].

Nadar and colleagues introduced zinc acetate solution into a water solution containing a mixture of $\alpha$-amylase, glucoamylase, PVP K-30, and mIM, preparing a dual-enzyme combination MOF. Its stability increased threefold, and after five consecutive reuse cycles, it retained as much as 52% of its residual activity [113]. In recent years, there has been increasing interest in synthesizing novel hybrid MOFs for enzyme immobilization by employing competitive ligands (amino acids, short peptide molecules, etc.) [114] along with traditional ligands. Competitive ligands inherently contain multiple coordination sites and various functional groups suitable for binding metal ions, enhancing the concentration of metal cations around enzymes to promote MOF nucleation. The addition of these competitive ligands can improve the interaction between enzymes and hybrid MOFs and the microenvironment in which they are located [115]. Chen et al. reported a novel strategy, amino acid-enhanced one-pot encapsulation, for the one-step preparation of enzyme–MOF composite materials. The key principle of this preparation method relies on the accelerated formation of a pre-nucleus cluster around proteins by the self-assembly of protein/PVP/cysteine (Cys). The coordination interaction between Cys and metal cations can accumulate metal cations, and Cys modification may expedite the formation of ligands and metal ion pre-nucleus clusters around the target protein, triggering MOFs to encapsulate these proteins [115].

### 3.1.2. Alkali Co-Precipitating Agents

Alkaline solutions also function as a type of co-precipitation agent. In the case of MOFs based on carboxylate salts, the organic ligand in an acidic state can be deprotonated by a base [116], facilitating the dissolution of the organic ligand and a direct reaction with metal and carboxylate salt groups. Alkaline solutions enhance the crystallization dynamics [117] by deprotonating the organic ligand. Gascón and colleagues added aluminum nitrate solution dropwise to a mixture containing $\beta$-glucosidase, 2-aminoterephthalic acid ($NH_2$-$H_2BDC$), and one of the three deprotonating agents (triethylamine (TEA), $NH_4OH$, or NaOH). This process resulted in the preparation of $\beta$-Glu@MIL-53-$NH_2$(Al), enhancing the enzyme loading in the MOFs. In the absence of deprotonating agents, the organic ligand cannot dissolve. The three deprotonating agents have varying abilities to dissolve the organic ligand, ranging from minutes to hours [109].

In the study by Liang et al., FCAT@MAF-7 was prepared by mixing zinc nitrate, 3-methyl-1,2,4-triazole (Hmtz), $NH_4OH$, and fluorescently labeled catalase (FCAT). Ammonia was necessary to deprotonate the Hmtz ligand. FCAT@MAF-7 exhibited continuous

enzymatic activity over 10 consecutive reactions, with no significant decrease in enzyme activity [118].

### 3.2. MOFs by Bionic Mineralization

Unlike the method involving the addition of co-precipitation agents, the biomimetic mineralization method involves the direct mixing of biomacromolecules with the organic ligands of MOFs. In this process, biomacromolecules act as the basic building blocks of MOFs, aiding in the nucleation of porous crystals and resulting in rapid crystallization around enzyme molecules. Hydrogen bonding and hydrophobic interactions between the organic ligands and enzyme molecules contribute to retaining the enzyme molecules within the MOF. The biomimetic mineralization method makes it possible for biomacromolecules to be fixed in situ during the synthesis process. This method does not require the addition of co-precipitation agents [108,119], and its preparation conditions are mild, making the process economically efficient. Enzymes are uniformly distributed throughout the entire MOF crystal [120]. As it avoids the use of co-precipitation agents, the biomimetic mineralization method for preparing enzyme–MOF biohybrids eliminates the risk of co-precipitation agents damaging enzymes at high temperatures. Therefore, materials prepared through biomineralization exhibit enhanced stability over a broader temperature range [107,121]. Liang et al. reported the first example of biomimetic mineralization of MOFs by mixing a zinc acetate solution with a solution containing mIM and enzymes at room temperature, resulting in a ZIF-8-based biohybrid material. This method demonstrated that PVP and alcohol were not necessary. Similar to co-precipitation, this process encapsulates the target biomolecules and preserves their biological activity by imparting high biotic, thermal, and chemical stability. Moreover, this method can be applied to other MOFs (HKUST-1, Eu/Tb-BDC, and MIL-88A), highlighting the versatility of this biomimetic mineralization approach [106]. Li et al. prepared Lipase@Bio-MOF by mixing zinc acetate, adenine (an organic ligand), and enzymes in an aqueous solution [122]. Lipase@Bio-MOF exhibited good catalytic activity and stability under high temperatures, alkaline conditions, and in the presence of metal ions. Additionally, it demonstrated excellent recyclability in the biodiesel production process, with no changes in morphology or crystal structure after three cycles. Furthermore, Zhang et al. utilized the biomimetic mineralization method to prepare Laccase@HKUST-1 biohybrids [123]. In terms of the catalytic efficiency, $K_{cat}/K_m$ for laccase@HKUST-1 improved by nearly four times that for the laccase. Research by Maddigan et al. indicated that the surface charge and chemical properties of proteins determine their ability to grow MOFs. Converting the basic residues on the protein surface into acidic or non-ionizable parts under standard conditions is a convenient strategy for promoting protein biomimetic mineralization [119].

The biomimetic mineralization process for preparing MOFs typically involves directly adding enzyme solutions to the encapsulation system. During this process, enzymes are usually stored in a solution containing various small molecules (such as NaCl, Tris, etc.), and these small molecules may impact the rate and morphology of MOF formation [124]. Pu and his team reported a method to prepare (*R*)-PEDH@ZIF-8 by mixing ultrasonically activated (*R*)-1-phenylethanol dehydrogenase ((*R*)-PEDH), zinc nitrate, mIM, and NaCl. They investigated the influence of NaCl on the preparation of enzyme–MOF and its catalytic performance. The $k_{cat}/K_m$ of the (*R*)-PEDH@ZIF-8 to (*R*)-1-phenylethanol was 1935 mM·s$^{-1}$, while the $k_{cat}/K_m$ of the free (*R*)-PEDH was 158 mM·s$^{-1}$, indicating that the (*R*)-PEDH@ZIF-8 (prepared with 0.1 M NaCl) has better substrate affinity than the free (*R*)-PEDH [125].

Mechanochemical processes, proven to be environmentally friendly alternatives to traditional solution-based processes, have been employed in the preparation of various MOFs [126]. The preparation of mechanochemical MOFs avoids the extensive use of organic solvents [127], and enzymes are more stable in powder form. He and colleagues successfully encapsulated thermophilic lipase (QLM) spontaneously in ZIF-8 through grinding, obtaining Lipase@ZIF-8. This composite was effectively applied to the kinetic

resolution of (*R*,*S*)-2-octanol, displaying good catalytic activity and enantioselectivity over 10 cycles of reactions [107].

In recent years, various enzymes have been individually encapsulated in MOFs, and the influence of MOF matrices on enzyme activity and stability has been studied. It is noteworthy that, so far, studies comparing the stability and activity of enzymes in MOFs with free enzymes in homogeneous media have mainly focused on individual enzymes rather than encapsulating multiple enzymes together [128]. Wu et al. reported a one-step, convenient synthesis of MOF nanocrystals containing multiple enzymes (Gox and HRP) in aqueous solution. The rigid structure and confinement of the MOF framework significantly enhanced the thermal stability of the encapsulated enzymes. Furthermore, it protected the encapsulated enzymes from protein hydrolysis and chelation effects [104].

MOFs have emerged as a rapid and effective method for enzyme immobilization, improving the catalytic stability and recyclability of enzymes. Despite the many advantages of MOFs in enzyme immobilization, several challenges remain. Some of these challenges include the following: (1) Regulate the composition of composite metals in MOFs to alter morphological structures, microscopic structures, catalytic activities, etc. (2) Improvements are needed in Achieve more uniform particle sizes and enhance the catalytic efficiency of MOFs. (3) Some MOFs tend to degrade under acidic conditions, thereby losing their protective effect on enzymes. Enhancing the "acid resistance" of materials is another issue that needs to be addressed. Addressing these challenges will contribute to the further development and optimization of MOF-based enzyme immobilization strategies.

## 4. Conductive Polymers

Conductive polymers (CPs) refer to polymers with a conjugated π-bonded long-chain structure. The polymer main chain is composed of alternating single and double bonds. CPs exhibit excellent conductivity, redox activity, environmental stability, and biocompatibility due to their conjugated structure. They are easily synthesized and can be used for enzyme immobilization [129]. Commonly used CPs include polypyrrole (PPy), poly(3,4-ethylene-dioxythiophene) (PEDOT), polyaniline (PANI), polythiophene (PTH), etc. [130]. A summary of one-step in situ enzyme immobilization in conductive polymer electropolymerization is presented in Table 3. The term "one-step in situ enzyme immobilization in conductive polymer electropolymerization" typically refers to the simultaneous presence of CP monomers and enzymes during the in situ polymerization process to form immobilized enzymes. In this process, the formation of CPs is usually achieved through electrochemical polymerization [131] or enzyme-catalyzed polymerization.

**Table 3.** One-step biosensors based on enzymes trapped inside a polymer.

| Polymer | Enzyme | Applications | Improved Performance | Ref. |
|---|---|---|---|---|
| PTh/Ppy/PANI | GOx | Glucose detection biosensor | Enhanced stability | [132] |
| NMPY | ChOx | Cholesterol biosensor | Enhanced charge transfer | [133] |
| PANI | PyOx | Glucose detection biosensor | Enhanced activity and stability | [134] |
| PANI | GOx/Ur | Glucose and urea enzymatic biosensors | Enhanced stability and reusability | [135] |
| Nafion®117 | FDH | Formaldehyde detection biosensor | Enhanced stability and reusability | [136] |
| Chitosan derivatives (CS-Fc) | GOD | Glucose detection biosensor | Enhanced electronic conductivity, electroactive surface area and electrochemical stability | [137] |

### 4.1. Immobilization by Electrochemical Polymerization of CP Monomers

Electrochemical polymerization is a method in which CP monomers and enzyme molecules are present in an electrolyte solution. By applying a voltage, CP monomers undergo oxidation to generate oligomers or polymers, which then co-deposit with enzyme molecules onto the electrode surface, achieving the immobilization of enzyme molecules [138,139]. The oxidation potential of polymers is always lower than that of monomers. Therefore, during the electrochemical polymerization process, monomers form radical cations on the electrode surface under the applied potential. These radical cations undergo dimerization through deprotonation, forming dimers that are subsequently oxidized again to form cations. Coupling reactions then occur between the cations and monomer radical cations, leading to chain propagation [140,141]. As a result, the main chain of the polymer carries a positive charge [142], which is neutralized by introducing anions from the electrolyte. Enzymes located near the electrode surface can be physically embedded into the polymer [141]. This has proven to be an effective way to capture enzymes in situ within organic polymers, providing a new means for the preparation of a variety of novel composite materials.

This method is the most direct approach for enzyme immobilization, offering the following distinct advantages [129,142]: (1) Preservation of enzyme activity and stability: immersing enzymes in conductive polymers helps maintain good enzyme activity and higher stability. (2) One-step process with fast immobilization: the entire process is completed in a single step, resulting in a fast immobilization rate. (3) Uniform distribution of immobilized enzymes: the immobilized enzymes are evenly distributed, independent of the geometric shape and size of the electrode. (4) Precise control of membrane thickness: the membrane thickness can be accurately controlled by regulating the amount of charge involved in the deposition steps. Deposition can be achieved using either constant potential or constant current methods [143].

Mello and colleagues employed an in situ electrochemical immobilization process by mixing a solution containing aniline monomers and enzymes (GOx or Ur) to prepare a polymerized aniline film embedded with enzymes. This resulted in glucose and urea biosensors incorporating immobilized GOx and Ur [135]. Additionally, Pramanik and colleagues developed a novel biosensing electrode by simultaneously electro-polymerizing pyrrole and co-depositing graphene oxide (gRGO) and cholesterol oxidase (ChOx). Compared to other reported cholesterol biosensors, this one-step-manufactured biosensor exhibited superior sensitivity, a broader linear response, and a lower detection limit. The excellent porosity of this composite material improved the efficiency of cholesterol diffusion, facilitating enzyme-catalyzed reactions [144,145].

PPy itself has a relatively low affinity for enzymes, leading to lower efficiency when using PPy as a CP monomer for enzyme immobilization. Enzymes are prone to release from the PPy matrix [146]. To address these issues, in recent years, various nanomaterials and other substances have been used in the enzyme immobilization process to enhance the efficiency, stability, and sensitivity of biosensors [147]. Lee and colleagues reported a one-step electrochemical method to prepare an electrode composed of PPy, polydopamine (PDA), and enzymes (such as glucose oxidase (GOx) or lactate oxidase (LOx)) (Figure 6). The addition of a small amount of PDA helped improve the efficiency of enzyme immobilization during the PPy polymerization process [138]. Rahim and collaborators used a one-step electro-polymerization approach to prepare a cholesterol nanobiosensor based on a Pt electrode, incorporating gold nanoparticles (AuNPs), cholesterol oxidase (COx), cholesterol esterase (CE), PPy, and $K_4Fe(CN)_6$. The addition of $Fe(CN)_6^{4-}$ and AuNPs enhanced the sensitivity of the cholesterol biosensor. The addition of $Fe(CN)_6^{4-}$ serves as a reducing electron mediator, which enhances the amperometric response of cholesterol by improving electron transfer between the enzyme and the electrode. The presence of AuNPs nanoparticles increases the surface area of the electrode, thereby reducing the impact of enzyme addition on the conductivity of PPy films. The synergistic effect of $Fe(CN)_6^{4-}$ and AuNPs significantly enhances the cholesterol reaction [148].

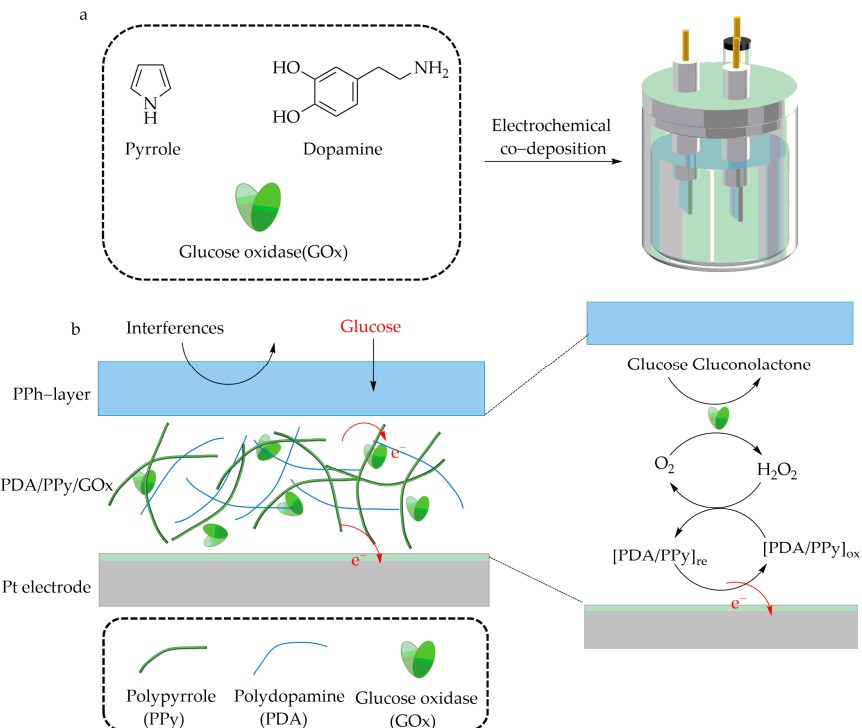

**Figure 6.** A schematic of polydopamine/polypyrrole/glucose oxidase (PDA/PPy/GOx) as a highly sensitive and stable amperometric glucose sensor. (**a**), One-pot chronopotentiometric co-deposition of GOx and PDA/PPy on electrodes. (**b**), The sensing mechanism of the PDA/PPy/GOx amperometric glucose sensor. [138].

### 4.2. Immobilization by Enzymatic Polymerization of CP Monomers

Another method to encapsulate enzymes within CPs is through enzyme-catalyzed polymerization. Enzymatic polymerization of CP monomers typically involves using a redox enzyme to catalyze the generation of a redox substance (such as $H_2O_2$), triggering a polymerization reaction [144]. This results in the formation of CPs, with the redox enzyme simultaneously being immobilized within the CPs.

Huang and colleagues utilized $H_2O_2$ oxidation to initiate the one-step oxidative polymerization of pyrrole-modified glucose oxidase (GOx), pyrrole-containing monomers (Py and 1-amino pyrrole (Py-NH$_2$)), and a cross-linker (Py-Py). This process led to the formation of a self-encapsulated nanoenzyme (n(GOx-PPy)). Compared to free GOx, n(GOx-PPy) exhibited excellent temperature stability and pH stability [149].

German and colleagues achieved the one-step polymerization of corresponding monomers (aniline, pyrrole, and thiophene) through enzyme-catalyzed polymerization, forming nanoparticles PANI/GOx, PPy/GOx, and PTh/GOx [150] (Figure 7). Additionally, Ramanavicius and collaborators immobilized GOx within PPy, enhancing the stability of GOx [132].

Hybrid materials containing inorganic nanoparticles (such as gold or silver nanoparticles) and CPs exhibit unique properties. The presence of oxygen or nitrogen elements in the polymer enhances its adsorption capability for inorganic nanoparticles, allowing the inorganic nanoparticles to form bonds with the conjugated diene segments of the CPs chains. German and colleagues reported the synthesis of PANI/AuNPs&GOx and Ppy/AuNPs&GOx nanocomposites through the introduction of AuNPs or chloroauric acid (HAuCl$_4$) to assist in the enzymatic catalysis of monomers aniline and pyrrole [151].

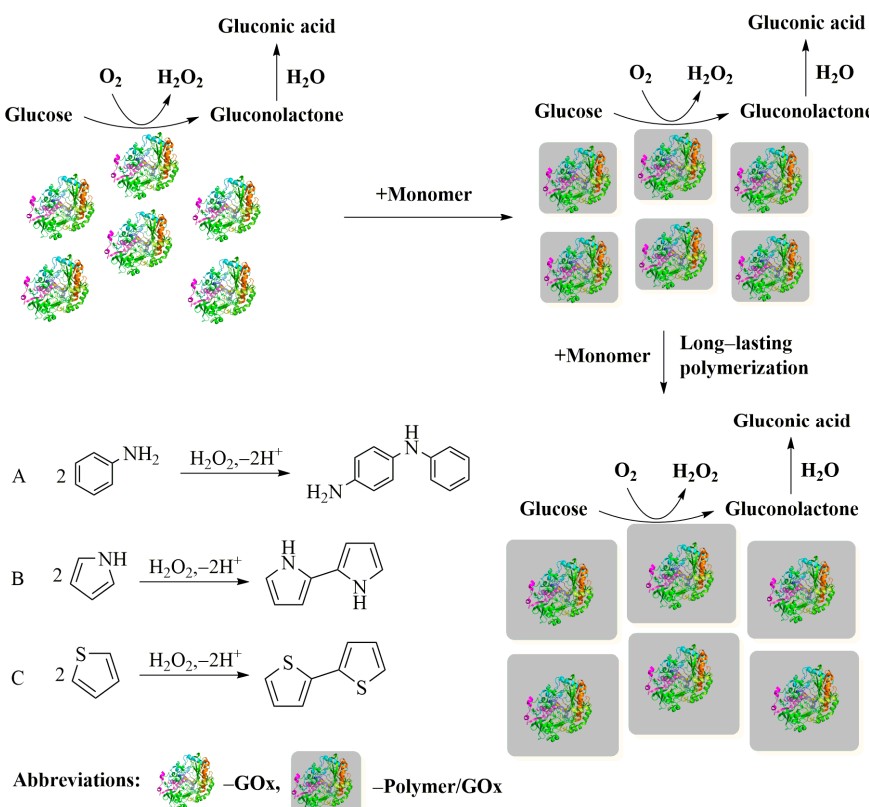

**Figure 7.** The formation of composite conducting polymer nanoparticles with embedded GOx by enzymatic polymerization and the schematic presentation of PANI (A), PPy (B), and PTH (C) polymerization mechanisms [150].

### 4.3. Other Strategies for Enzyme Immobilization by Electrochemistry

Non-conductive polymers are an emerging matrix for immobilizing biomolecules [152]. Studies have indicated that non-conductive polymers can also be prepared for enzyme immobilization through a one-step electrodeposition strategy. Films formed by non-conductive polymers are typically thinner than CP layers, resulting in faster substrate and product diffusion rates. Additionally, non-conductive polymers can prevent interference from electroactive substances in samples, offering advantages such as high sensitivity, rapid response time, and good reproducibility [153].

The biopolymer composite material composed of enzymes, Nafion, and CPs exhibits excellent stability. Nafion, acting as a binder, ensures that enzymes adsorb extensively in a manner conducive to bioelectrocatalysis and prevents enzymes from detaching from the electrode surface. Semenova and her colleagues have reported a novel one-step synthesis method that relies on the co-deposition of oxidases (such as glucose oxidase or alcohol oxidase), Nafion®117, and palladium nanoparticles (Pd-NPs) through the electrochemical deposition of a phosphate multi-electrolyte. Biosensors prepared through this method exhibit significantly enhanced mechanical stability with virtually no loss of enzyme activity [154].

Silicon dioxide ($SiO_2$) films have attracted widespread interest as a material for enzyme immobilization due to their optical transparency and ease of preparation. Jia et al. introduced a one-step electrochemical method on a platinum electrode to fabricate a glucose oxidase (GOD) biomaterial encapsulated within a porous sol-gel silicon dioxide matrix. The GOD encapsulated by this method exhibited superior bioactivity and stability compared to electrodes produced through the physical adsorption of GOD. The enzyme activity remained virtually unchanged after storage for 20 h and retained 60% activity after 120 h of storage [155].

Ruthenium-modified complexes can also serve as matrices for immobilized enzymes. Otero and colleagues employed an electrochemical method to one-step immobilize diaphorase and ruthenium-based polymers on a nanoporous gold electrode (NPG), creating an electrochemical sensor for NADH. Even after storage at 4 °C for 24 h, the catalytic reaction rate remained at 63% [156].

One-step in situ enzyme immobilization in polymer electropolymerization strategies has found wide application in the fabrication of biosensors for detecting glucose, ethanol, cholesterol, and xanthine. Due to its simple and rapidly controllable preparation steps, as well as the uniform distribution of enzymes, this strategy exhibits unique advantages in the field of biosensors. However, challenges such as the limited variety of CPs and the high cost of non-conductive polymers still constrain the application of one-step in situ enzyme immobilization in polymer electropolymerization. Future research directions should focus on expanding the types of CPs or incorporating other nanomaterials to address these limitations.

## 5. Summary and Outlook

We have reviewed and discussed three strategies for enzyme immobilization that are achieved simultaneously during the synthesis of nanocarriers. These strategies include organic–inorganic hybrid nano-flowers (HNFs), metal–organic frameworks (MOFs), and conductive polymers (CPs) as nanocarriers for enzyme immobilization. These three strategies avoid cumbersome preparation methods, maintain mild preparation conditions, and simplify the preparation process and workflow for enzyme immobilization. However, these three enzyme immobilization methods still have certain issues. For instance, HNFs exhibit poor stability. Although some studies have combined them with other immobilization methods or incorporated additional substances during the preparation process to regulate biomimetic mineralization, not all enzymes are suitable for this approach, limiting its applicability. Future efforts should focus on enzyme modification or changing the anions used in HNF preparation to expand the range of enzymes suitable for HNF preparation. MOFs, based on coordination bonding for precipitation, suffer from inadequate stability due to their single coordination bond. Moreover, the precipitation of MOFs not only requires consideration of enzyme electronegativity but also the strength of coordination interactions between central metal ions and organic ligands, which can affect MOFs' stability. Future directions should involve finding new co-precipitants to increase coordination interactions and promote MOF nucleation, thereby enhancing their stability. One-step in situ enzyme immobilization in a conductive polymer electropolymerization strategy based on physical embedding within polymers may significantly impact enzyme activity. Future research should focus on expanding the types of conductive polymers to improve enzyme stability. Currently, there are only these three strategies mentioned in this paper for achieving enzyme immobilization at the nanoscale during carrier synthesis, but they have shown significant potential. Future efforts should aim to improve the technical aspects of these strategies while also exploring and developing new immobilization strategies. This approach has the potential to leverage the advantages of simplicity, mild conditions, and other benefits, ultimately achieving the goal of green, clean, and rapid enzyme immobilization.

**Author Contributions:** Conceptualization, P.L. and X.H.; writing—original draft preparation, X.H.; writing—review and editing, P.L.; project administration, X.C.; funding acquisition, X.C. All authors have read and agreed to the published version of the manuscript.

**Funding:** This research was funded by the National Key Research and Development Program of China (2021YFC2101000).

**Conflicts of Interest:** The authors declare no conflicts of interest.

## Abbreviations

| Abbreviations | Full Name | Abbreviations | Full Name |
|---|---|---|---|
| HNFs | Organic–inorganic hybrid nanoflowers | HRP | Horseradish peroxidase |
| MOFs | Metal–organic frameworks | GOx | Glucose oxidase |
| ZIF-8 | Zeolitic imidazolate framework-8 | Ur | Urease |
| ZIF-90 | Zeolitic imidazolate framework-90 | Cyt c | Cytochrome c |
| MIL-53 | Materials of Institut Lavoisier-53 | Cys | Cysteine |
| MIL-88 | Materials of Institut Lavoisier-88 | $\beta$-Glu | $\beta$-glucosidase |
| MIL-100 | Materials of Institut Lavoisier-100 | PyOx | Pyranose oxidase |
| $NH_2$-$H_2BDC$ | 2-Aminoterephthalic acid | FDH | Formate dehydrogenase |
| Eu/Tb-BDC | $Eu_2(1,4\text{-}BDC)_3(H_2O)_4$/$Tb_2(1,4\text{-}BDC)_3(H_2O)_4$ | CPO | Chloroperoxidase |
| MIL-88A | Materials of Institut Lavoisier-88A | QLM | Thermophilic lipase |
| MAF-7 | Metal azolate framework-7 | gRGO | Graphene oxide |
| HKUST-1 | Hong Kong University of Science and Technology-1 | ChOx | Cholesterol oxidase |
| PDA | Polydopamine | LOx | Lactate oxidase |
| EDTA | Ethylenediaminetetraacetic acid | COx | Cholesterol oxidase |
| ABTS | 2, 2′-Azino-bis(3-ethylbenzothiazoline-6-sulfonic acid) | FCAT | Fluorescently labeled catalase |
| $HAuCl_4$ | Chloroauric acid | GOD | Glucose oxidase |
| PVP | Polyvinylpyrrolidone | CE | Cholesterol esterase |
| PVP K-30 | Polyvinylpyrrolidone K-30 | Ge | Gelatin |
| CNTs | Carbon nanotubes | CS | Chitosan |
| MNPs | Magnetic nanoparticles | CS-Fc | Chitosan derivatives |
| AuNPs | Gold nanoparticles | CaP | Calcium phosphate |
| Pd-NPs | Palladium nanoparticles | SPase | Sucrose phosphorylase |
| (R)-PEDH | (R)-1-phenylethanol dehydrogenase | $\omega$-ta | Omega-transaminase |
| $K_4Fe(CN)_6$ | Potassium ferrocyanide | PLP | 5′-Pyridoxal phosphate |
| NADH | Nicotinamide adenine dinucleotide | $SiO_2$ | Silicon dioxide |
| CPs | Conductive polymers | mIM | 2-Methylimidazole |
| PPy | Conductive polymers | Hmtz | 3-Methyl-1,2,4-triazole |
| PEDOT | Poly(3,4-ethylene-dioxythiophene) | TEA | Triethylamine |
| PANI | Polyaniline | NPG | Nanoporous gold electrode |
| PTH | Polythiophene | Py-$NH_2$ | Amino pyrrole |

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
