# Peer review of "Recent Advances in the Strategies of Simultaneous Enzyme Immobilization Accompanied by Nanocarrier Synthesis"

_applsci, doi:10.3390/app14093702_

Round 1

Reviewer 1 Report

Comments and Suggestions for Authors

The manuscript of the review article by Chu et al. presents the state of the art of various in-situ enzyme immobilization strategies using a few rather arbitrarily selected types of nanomaterials. I would ask the authors to consider the following issues:

- the title is too general and should be specified, as the manuscript describes selected types of nanomaterials with rather special properties

- please check the list of references carefully. There are, for example, duplicated items (e.g., ref. 10 = ref. 11)

- line 34-57: the description of the enzyme immobilization process is very general, resembling an extract from an academic handbook rather than an introduction to a review article on a relatively narrow topic

- line 15: this sentence is unclear, especially the use of the word "electrochemical" in combination with HNFs and MOFs

- line 66: the manuscript describes very specific and only selected types of nanomaterials, so the use of the term "representative" is controversial. Please provide a better justification for why the selected examples can be considered representative. What distinguishes HNFs from other anisotropic nanomaterials?

- line 112: it is worth mentioning earlier the key role of phosphates in the formation and structure of HNFs

- there are quite a few figures in the manuscript; for example, the paragraph from line 118 could have a scheme

- the Authors often refer to works in which it was found that immobilization improved enzyme activity (e.g., several times - see line 183, and others). This is a rather general statement, and it is worth specifying each time what parameter was improved. How the concentrations of immobilized and native enzymes were determined (because only then would such a direct quantitative comparison of activity make sense)

- the overwhelming number of abbreviations makes the manuscript difficult to understand. At the very least, I think it's worth carefully examining whether each of them (other than the commonly known ones) is explained the first time they are used. A list of abbreviations would probably also be a good idea.

- Fig. 1 – text within the blue labels is completely unreadable

- Table 2: Its presence is difficult to understand. Unless it is supplemented with at least a list of references - examples of applications or extended considerations of the effect of structure on the properties of MOFs, drawing formulas (which anyone can quickly find based on the name of the compound) seems unnecessary

- why was only PVP distinguished as a coprecipitating agent? What about the others? The selection of only PVP should be better justified

- what are the mechanisms of MOFs precipitation? It is worth adding a short explanation of this issue

- line 422: please verify reference number [1000].

- line 538-540: the statement that the additives improved the sensitivity of the cholesterol biosensor is obvious and vague... Please provide information (even in a few words) on what was the essence of the improvement and what kind of process influenced it

Line 582: this sentence needs to be clarified. Contextual examples of the materials described need to be included. It is even difficult to clearly state what is a biopolymer - whether the matrix or the enzyme.

Author Response

Dear Reviewer,

We appreciate you very much for your positive and constructive comments on our manuscript. We have fully revised our manuscript and have addressed all comments. The detailed revisions are listed below and highlighted in the revised manuscript.

  • The title is too general and should be specified, as the manuscript describes selected types of nanomaterials with rather special properties.

Response: Thanks a lot for reviewer’s suggestion. We have revised the title to "Recent advances in the strategies of simultaneous enzyme immobilization accompanied by nanocarrier synthesis".

  • Please check the list of references carefully. There are, for example, duplicated items (e.g., ref. 10 = ref. 11).

Response: Thanks for your careful checks. We are sorry for our carelessness. The duplicated reference has been removed (Line 788 in the revised manuscript).

  • Line 34-57: the description of the enzyme immobilization process is very general, resembling an extract from an academic handbook rather than an introduction to a review article on a relatively narrow topic.

Response: Thank you for your suggestion. We redescribe the method and process of enzyme immobilization for lines 34–57 (Line 84-98 in the revised manuscript).

  • Line 15: this sentence is unclear, especially the use of the word "electrochemical" in combination with HNFs and MOFs.

Response: Thank you for the detained review. We have changed electrochemical to conductive polymers (Line 26 in the revised manuscript).

  • Line 66: the manuscript describes very specific and only selected types of nanomaterials, so the use of the term "representative" is controversial. Please provide a better justification for why the selected examples can be considered representative. What distinguishes HNFs from other anisotropic nanomaterials?

Response: Thank you for your suggestion. In this review, we focus on those strategies for simultaneous enzyme immobilization accompanied by carrier synthesis. Those strategies have advantages such as mild reaction conditions, simple preparation processes, and minimal impact on enzymes, so they can complete one step enzyme immobilization in nanocarrier synthesis process. In this review, we reviewed three strategies that conform to the above principles, including organic-inorganic nanoflowers (HNFs), metal-organic frameworks (MOFs), and conductive polymers (CPs). We focus on discussing the principles, preparation methods, immobilization effects, and limitations of these three methods. Apart from HNFs, MOFs and CPs, we did not find any other strategies or anisotropic materials suitable for simultaneous enzyme immobilization in carrier synthesis processes. Therefore, this paper did not also address the differences between nanoflowers and other anisotropic materials.

  • Line 112: it is worth mentioning earlier the key role of phosphates in the formation and structure of HNFs.

Response: Thank you for your suggestion. Currently, no one has specifically discussed the role of phosphate in the preparation process of HNFs. On the basis of our understanding, we have added a new paragraph by outlining the crucial role that phosphates play in the synthesis and structure of HNFs. (Line 160-170 in the revised manuscript).

  • There are quite a few figures in the manuscript; for example, the paragraph from line 118 could have a scheme.

Response: We think this is an excellent suggestion. For simplicity of understanding, we have repositioned Table 1 (Line 178 in the revised manuscript).

  • The Authors often refer to works in which it was found that immobilization improved enzyme activity (e.g., several times - see line 183, and others). This is a rather general statement, and it is worth specifying each time what parameter was improved. How the concentrations of immobilized and native enzymes were determined (because only then would such a direct quantitative comparison of activity make sense).

Response: Thanks for your suggestion. In cases where the enzyme activity is observed to increase after immobilization, specific changes in parameters are further explained and detailed in the review if there are specific reports in the literature (Line 244-247, 259-260, 415-418, 499-501, 514-517 in the revised manuscript). Before immobilizing enzymes, the free enzyme protein content is usually measured using the Bradford method or the Pierce™ BCA protein assay. After immobilization, the concentration of residual protein in the supernatant is determined using the same method to calculate the immobilized enzyme content. For activity comparison, it is common to calculate the activity of the immobilized enzyme by comparing it with an equivalent amount of free enzyme based on protein content.

  • The overwhelming number of abbreviations makes the manuscript difficult to understand. At the very least, I think it's worth carefully examining whether each of them (other than the commonly known ones) is explained the first time they are used. A list of abbreviations would probably also be a good idea.

Response: We think this is an excellent suggestion. To make it easier to understand, we've created a new list of abbreviations (Line 754 in the revised manuscript).

  • 1 – text within the blue labels is completely unreadable.

Response: We sincerely thank the reviewer for careful reading. We have modified Fig 1. (Line 238 in the revised manuscript).

  • Table 2: Its presence is difficult to understand. Unless it is supplemented with at least a list of references - examples of applications or extended considerations of the effect of structure on the properties of MOFs, drawing formulas (which anyone can quickly find based on the name of the compound) seems unnecessary.

Response: We sincerely thank the reviewer for careful reading. We have removed Table 2.

  • Why was only PVP distinguished as a co-precipitating agent? What about the others? The selection of only PVP should be better justified.

Response: We sincerely appreciate the valuable comments. In this review, we extensively discuss PVP and alkalis as co-precipitating agent for MOFs (Line 403,441 in the revised manuscript). In the preparation process of MOFs, co-precipitating agent typically include various acids, alkalis, and surfactants. Although there are many surfactants that can be used for MOF synthesis, our literature review revealed that in strategies for immobilizing enzymes during MOFs synthesis, only alkalis and PVP are commonly used as co-precipitating agent. Therefore, our paper only discusses alkalis and PVP as co-precipitating agent for MOFs preparation, and other co-precipitating agent are not within the scope of our discussion.

  • What are the mechanisms of MOFs precipitation? It is worth adding a short explanation of this issue.

Response: Thanks for your good comments. We have added a new description of the mechanism of precipitation of MOFs to the review (Line 344-350 in the revised manuscript).

  • Line 422: please verify reference number [1000].

Response: We are sorry for our carelessness. We have changed the reference to the correct serial number (Line 498 in the revised manuscript).

  • Line 538-540: the statement that the additives improved the sensitivity of the cholesterol biosensor is obvious and vague... Please provide information (even in a few words) on what was the essence of the improvement and what kind of process influenced it.

Response: We sincerely appreciate the valuable comments. We have gone into greater depth regarding the addition of compounds to make cholesterol sensors more sensitive. (Line 627-632 in the revised manuscript).

  • Line 582: this sentence needs to be clarified. Contextual examples of the materials described need to be included. It is even difficult to clearly state what is a biopolymer - whether the matrix or the enzyme.

Response: We think this is an excellent suggestion. The biopolymer has been thoroughly explained (Line 678-681 in the revised manuscript).

Once again, thank you very much for your valuable comments and suggestions. We would be glad to response to any further questions and comments that you may have.

Thank you and best regards.

Sincerely,

Xiaohe Chu

Reviewer 2 Report

Comments and Suggestions for Authors

In the manuscript “Recent advances in in-situ immobilization of enzyme based on 2 nanomaterials”, authors made an attempt to generalize information about various in-situ enzyme immobilization strategies based on nanomaterials.

However, some aspects can be improved:

1.       Please, as clearly as possible, state the problem of the current review in the last paragraph of the introduction section. The idea of the review is not clear.

2.       Please cite the reference which corresponds Figure 1, Figure 2, Figure 3, Figure 4, Figure 5, figure 6 and 7.

3.       For better understanding, it is highly recommended to make generalized classification methods for organic-inorganic hybrid nanoflowers, MOFs, and CPs in graphic form, maybe a scheme.

4.       Probably the Summary section should be reconsidered. So far, it doesn’t report any original knowledge or outcomes.

Author Response

Dear Reviewer,

We are very grateful for your professional suggestions on our article. We have considered the comments carefully and tried our best to revised the manuscript accordingly. Our responses are given in a point-by-point manner below. The detailed revisions are listed below and highlighted in the revised manuscript.

  • Please, as clearly as possible, state the problem of the current review in the last paragraph of the introduction section. The idea of the review is not clear.

Response: We think this is an excellent suggestion. We have re-written the introduction section (Line 84-118 in the revised manuscript).

  • Please cite the reference which corresponds Figure 1, Figure 2, Figure 3, Figure 4, Figure 5, figure 6 and 7.

Response: Thank you for the suggestion. We have referenced Figure 2, Figure 3, Figure 4, Figure 5, Figure 6, figure 7 and 8 (Line 239, 250, 296, 422, 541, 635, 658 in the revised manuscript).

  • For better understanding, it is highly recommended to make generalized classification methods for organic-inorganic hybrid nanoflowers, MOFs, and CPs in graphic form, maybe a scheme.

Response: We think this is an excellent suggestion. We have created a flowchart for the generalized classification of HNFs, MOFs, and CPs based on this review (Line 119 in the revised manuscript).

  • Probably the Summary section should be reconsidered. So far, it doesn’t report any original knowledge or outcomes.

Response: We sincerely appreciate the valuable comments and we've modified the Summary section (Line 725-752 in the revised manuscript).

Again, thank you for giving us the opportunity to strengthen our manuscript with your valuable comments and queries. We have worked hard to incorporate your feedback and hope that these revisions persuade you to accept our submission.

Thank you and best regards.

Sincerely,

Xiaohe Chu

Reviewer 3 Report

Comments and Suggestions for Authors

The paper entitled “Recent advances in in-situ immobilization of enzyme based on nanomaterials” reviews the data existing in the field of in-situ enzymes immobilization. It offers valuable information on the process carried out by using nanomaterials and strategies such as organic-inorganic nanoflowers, metal-organic frameworks and electrochemical. These three methodologies are discussed and summarized and the exposed ideas are sustained with examples of multiple published works.

The approached subject is interesting especially in the actual context of the continuous efforts of development of rapid, environmentally-friendly and reliable methods in the studied domain.

The authors will find bellow some corrections and adjustments that should be addressed.

-          Since the paper is a review, it would be appropriate to specify what was the bibliographic research methodology applied for obtaining the information.

-          What databases were consulted?

-          What was the time period chosen?

-          What were the keywords chosen for the research?

-          What were the criteria on the basis of which some references were included/excluded in/from the review?   

-          Why these specific three strategies for in-situ immobilization of enzymes were chosen to be presented? What other methods (besides those already discussed and the classical ones) can be used? With what results?

-          Since the title of the review is rather general, in addition to the use in biocatalysis (as mentioned in the Abstract and developed later in the paper), what other areas can benefit of the discussed immobilization strategies? With what results?  

Comments on the Quality of English Language

Minor editing of English language required

Author Response

Dear Reviewer,

We feel great thanks for your professional review work on our article. As you are concerned, there are several problems that need to be addressed. According to your nice suggestions, we have made extensive corrections to our previous draft, the detailed corrections are listed below.

  • What databases were consulted?

Response: Thank you for the question. The main databases we have chosen are PubMed、Elsevier、ACS Publications、Springer nature、Web of Science and others.

  • What was the time period chosen?

Response Thank you for the question. The vast majority of the literature is within the last five years.

  • What were the keywords chosen for the research?

Response: Thank you for the question. The main keywords we have chosen are: MOF&enzyme、HNF&enzyme、electrochemically&enzyme immobiliazation、in-suit&enzyme&immobilized、nanomaterials、one step&enzyme&immobilized and conductive polymers&enzyme .

  • What were the criteria on the basis of which some references were included/excluded in/from the review?

Response: Thank you for the question. The selection of references was based on three criteria: 1. the year of publication, specifically if the reference falls within the last five years; 2. the literature's description of the enzyme immobilization technique as a means of achieving nanoscale immobilization of the enzyme during carrier synthesis; and 3. the literature's research content as a representative sample of this class of immobilized enzyme preparation strategies.

  • Why these specific three strategies for in-situ immobilization of enzymes were chosen to be presented? What other methods (besides those already discussed and the classical ones) can be used? With what results?

Response: Thank you for the question. We have revised the title to "Recent advances in the strategies of simultaneous enzyme immobilization accompanied by nanocarrier synthesis". In this review, we focus on those strategies for simultaneous enzyme immobilization accompanied by carrier synthesis. Those strategies have advantages such as mild reaction conditions, simple preparation processes, and minimal impact on enzymes, so they can complete one step enzyme immobilization in nanocarrier synthesis process. In this review, we reviewed three strategies that conform to the above principles, including organic-inorganic nanoflowers (HNFs), metal-organic frameworks (MOFs), and conductive polymers (CPs). We focus on discussing the principles, preparation methods, immobilization effects, and limitations of these three methods. Other immobilization strategies typically involve immobilizing enzymes on commercial carriers or pre-prepared carriers, such as common methods like COFs, mesoporous silica, functionalized magnetic nanoparticles, and others. Therefore, we have chosen to focus on these three strategies in our discussion.

  • Since the title of the review is rather general, in addition to the use in biocatalysis (as mentioned in the Abstract and developed later in the paper), what other areas can benefit of the discussed immobilization strategies? With what results?

Response: Thank you for the question. We have revised the title to " Recent advances in the strategies of simultaneous enzyme immobilization accompanied by nanocarrier synthesis", the scope of this paper has been narrowed down. The enzyme immobilization strategies outlined in the article are applicable in the field of biocatalysis, and there are many other application areas beyond biocatalysis. For example, they can also be used for environmental protection and the preparation of biosensors. In this review, Tables 1, 2, and 3 respectively summarize the applications and improved performance of enzyme immobilization based on HNFs, MOFs, and CPs (Line 178, 389, 556 in the revised manuscript).

We deeply appreciate your all the valuable comments and suggestions, and look forward to hearing from you regarding our submission. We would be glad to response to any further questions and comments that you may have.

Thank you and best regards.

Sincerely,

Xiaohe Chu

Round 2

Reviewer 1 Report

Comments and Suggestions for Authors

Thank you for the work you have done in preparing the responses and revised manuscript. In my opinion, the changes made have positively affected the overall quality of the manuscript and I can recommend it for publication in its current form. 

Author Response

Dear Reviewer,

We appreciate your helpful suggestions, which helped to improve the revised manuscript's readability and scientific soundness.

Thank you and best regards.

Sincerely,

Xiaohe Chu

Reviewer 3 Report

Comments and Suggestions for Authors

The paper entitled “Recent advances in the strategies of simultaneous enzyme immobilization accompanied by nanocarrier synthesis” (with the initial title “Recent advances in in-situ immobilization of enzyme based on nanomaterials”) reviews the data existing in the field of in-situ enzymes immobilization.

I congratulate the authors for taking into account the suggestions of the reviewers and for their efforts in bringing the necessary clarifications and making the requested changes. The work is much improved compared to the initial version and can be accepted for publication after minor revision. This revision concerns the methodology used for selecting the conclusive references included in the review paper. It is recommended to add the information provided as answer to the first round of the peer-review process in a new section of the manuscript.

Author Response

Dear Reviewer,

We appreciate you very much for your positive and constructive comments on our manuscript. We have fully revised our manuscript and have addressed all comments. The detailed revisions are listed below and highlighted in the revised manuscript.

  • This revision concerns the methodology used for selecting the conclusive references included in the review paper. It is recommended to add the information provided as answer to the first round of the peer-review process in a new section of the manuscript.

Response: Thanks for your suggestion. We have added the methodology used for selecting the conclusive references to the manuscript (Line 75-95 in the revised manuscript).

Once again, thank you very much for your valuable comments and suggestions. We would be glad to response to any further questions and comments that you may have.

Thank you and best regards.

Sincerely,

Xiaohe Chu
